# Pre- and Post-Zygotic Barriers Contribute to Reproductive Isolation and Correlate with Genetic Distance in *Cucumis*

**DOI:** 10.3390/plants12040926

**Published:** 2023-02-17

**Authors:** María Ferriol, Unzué Simó, Carme J. Mansanet, Alejandro Torres, Belén Picó, Antonio J. Monforte, Carlos Romero

**Affiliations:** 1Instituto Agroforestal Mediterráneo, Universitat Politècnica de València, Camino de Vera s/n, 46022 Valencia, Spain; 2Instituto de Biología Molecular y Celular de Plantas, Consejo Superior de Investigaciones Científicas—Universitat Politècnica de València, c/Fausto Elio s/n, 46022 Valencia, Spain; 3Instituto Universitario de Conservación y Mejora de la Agrodiversidad Valenciana, Universitat Politècnica de València, c/Fausto Elio s/n, 46022 Valencia, Spain

**Keywords:** *Cucumis*, interspecific reproductive barriers (IRBs), unilateral cross-incompatibility, fluorescence microscopy, pollen tube growth, reproductive isolation

## Abstract

Hybridization between *Cucumis* species, including cultivated melon (*C. melo*), is hampered by Interspecific Reproductive Barriers (IRBs). However, the nature of IRBs in *Cucumis* is largely unknown. This study explores locations, timing, and contribution to reproductive isolation (RI) of pre- and post-zygotic IRBs in *Cucumis*. To do this, we assessed crossability among *Cucumis* African wild species and *C. melo* at the pre-zygotic level by visualizing pollen tubes under fluorescence microscopy and, post-zygotically, by evaluating fruit/seed set and F_1_ hybrid fertility. Genetic distances among *Cucumis* species were inferred from Genotyping-by-Sequencing, and its correlation with RI stages was analyzed. Observed pre- and post-zygotic IRBs included pollen tube arrest, fruit set failure, and hybrid male sterility. Unilateral cross-incongruity/incompatibility (UCI) was detected in some hybridizations, and dominant gene action is suggested for pistil-side UCI in interspecific F_1_ hybrids. Notably, the allotetraploid *C. ficifolius* was very fertile as a seed parent but infertile in all reciprocal crosses. Contribution to RI was found significant for both pre- and post-zygotic IRBs. Additionally, a significant positive correlation was detected between genetic distance and pre- and post-zygotic RI stages. Interestingly, UCI offers an accessible system to dissect the genetics of IRBs in *Cucumis*, which may facilitate the use of wild relatives in breeding.

## 1. Introduction

The *Cucumis* genus belongs to the Cucurbitaceae family (Cucurbitales order within the Rosids clade) [1] and groups 66 species, including some of major economic importance such as cultivated melon (*C. melo* L.) and cucumber (*C. sativus* L.) [2]. *Cucumis* species are herbaceous and, generally, creeping and climbing plants with a shallow radicular system. Most of them are monoecious and frequently exhibit solitary male and female flowers. These latter have an inferior ovary and produce fruits with a wide range of colors, sizes, and shapes (round, oval, and/or elongated) [3].

Regarding the reproductive behavior, self-incompatibility (SI) has not been recorded in any *Cucumis* species [4] though monoecy and, to a lesser extent, andromonoecy and dioecy are widely extended in this genus [2]. Furthermore, evidence supports that a common ancestor of Cucurbitaceae, probably underwent the two major transitions among sexual systems: self-compatibility (SC) from SI and sexual dimorphism from hermaphroditism [5,6,7]. SC and monoecy operate in opposite directions by promoting self- and cross-fertilization, respectively, and their evolutionary spread may rely on the fitness consequences of self- and out-crossing, among other factors [5,8].

At the interspecific level, early studies (as reviewed by Chen and Zhou [9]) reported that hybridization was feasible between some wild African *Cucumis* species grouped into the *Aculeatosi* section but rather rare between different sections [10,11]. The observed crossability patterns were roughly similar among all these studies, and discrepancies may be attributable to differences in the experimental conditions and the genotypes used [4,10,12,13]. For instance, *C. dipsaceus* Ehrenb. ex Spach crossed successfully as a female parent with *C. zeyheri* Sond. and *C. myriocarpus* Naudin, according to Singh and Yadaba [13] but not to den Nijs and Visser [10]. Notably, cultivated melon (*C. melo*) has not been consistently crossed with any other *Cucumis* species to date. Some early attempts with *C. sagittatus* Wawra and Peyr. [14] and *C. metuliferus* E. Mey. ex Naudin [15] produced only embryos or could not be successfully replicated, respectively. Much later, Matsumoto et al. [16] were able to enhance melon pollen tube performance onto *C. anguria* L. pistils by pollinating at high temperatures, overcoming pre-zygotic barriers, and obtaining fruits but again with non-viable seeds. At odds with melon, a successful interspecific cross was performed between the cultivated cucumber (*C. sativus*) and *C. hystrix* Chakrav, two phylogenetically closely related Asiatic species with different chromosome numbers [17]. This cross generated the new taxa *Cucumis* × *hytivus* J.F. Chen and J.H. Kirkbr., that may be useful as bridge species for transferring agronomically interesting traits [18]. Overcoming interspecific barriers to using wild germplasm in breeding programs is also a relevant issue in other important crop genera, such as *Solanum* which includes tomato and potato clades. In fact, deep knowledge of the Solanum IRBs paved the way for this purpose over more than a decade, particularly by using unilateral-incompatibility-related factors [19].

However, the nature of the Interspecific Reproductive Barriers (IRBs) operating in *Cucumis* is largely unknown. At the pre-zygotic level, Kishi and Fujishita [12] already observed that, compared with selfing, pollen tubes grew much slower in incompatible crosses and rarely reached the ovules. In selfings and compatible crosses, fertilization takes place between 24 and 72 h after pollination (depending on the environmental conditions, especially the temperature), and the arrest of pollen tubes at the stigma, style, or ovary may also be observed at this time span [4,12]). Moreover, in compatible crosses, thin pollen tubes grow normally, showing the characteristic callose plugs along their length, but in incompatible crosses, they are usually found to be distorted and filled with callose [4]. Despite this, pollen tubes have been observed to enter the ovules in some incompatible crosses. For instance, Kishi and Fujishita [12] found that in *C. dipsaceus* × *C. melo* and *C. dipsaceus* × *C. figarei* Delile ex Naudin crosses, a few slowly growing pollen tubes entered into the ovules, but zygote cells did not divide and endosperms finally aborted. These results suggested that post-zygotic barriers related to embryogenesis and seed formation were involved in cross-incompatibility in *Cucumis*. Additionally, hybrid seeds are sometimes non-viable, and hybrids frequently exhibit male and/or female sterility [10]. Thus, IRBs have presumably contributed along with other factors (geographic, etc.) to the reproductive isolation of *Cucumis* species.

Accordingly, early attempts to define a *Cucumis* phylogeny were mainly based on crossability data but also on pollen fertility [14], chromosome pairing [13], or meiotic analysis [10]. The first comprehensive phylogeny for *Cucumis* based on morphology and biosystematic data (chromosome number, crossability, isozymes, etc.) was established by Kirkbride [3]. This classification distinguished two subgenera; *Melo* originated in Africa with 30 spp. (chromosome number 2*n* = 24) that included the cultivated *C. melo* and *Cucumis* originated in Asia with two spp. (chromosome number 2*n* = 14) including *C. sativus* (cucumber). Since then, studies have focused on the analysis of DNA sequences to understand *Cucumis* phylogeny. García-Más et al. [20] used nuclear markers, such as Simple Sequence Repeats (SSR) and sequence variations in internal transcribed spacer regions (ITS1 and ITS2) of the ribosomal RNA genes, for the phylogenetic analysis and results roughly supported Kirkbride’s [3] classification. However, later molecular phylogenetic studies based on DNA sequences from chloroplast genes (*rbcL*, *matK*, *rpl20-rps12*, *trnL*, and *trnL-F*), nuclear ITS, and plastid *trnS-trnG* regions indicated that species from close related genera (*Cucumella*, *Dicaelospermum*, *Mukia*, *Myrmecosicyos*, and *Oreosyce*) should be incorporated into *Cucumis* [21,22]. Furthermore, Sebastian et al. [2] performed an exhaustive phylogenetic study in *Cucumis* by analyzing nuclear and chloroplastic DNA diversity again. The authors concluded that *Cucumis* species are mainly distributed throughout Africa, Asia, and Australia and suggested that the probable ancestor of the cultivated melon was located in India or Australia. Altogether, these last works [2,21,22] question the *Cucumis* subdivision established by Kirkbride [3]. More recently, Endl et al. [23] stated that melon was probably domesticated at least twice in Africa and Asia, and modern melon cultivars can be traced back to two lineages: *Cucumis melo* subsp. *melo* restricted to Asia and *C. melo* subsp. *meloides* restricted to Africa.

Despite all these studies, the speciation process in *Cucumis* and, particularly, the role played by IRBs are far from being known. As an attempt to shed light on this issue, here we have characterized a wide range of pre- and post-zygotic barriers, showing that both IRB types contribute significantly to the reproductive isolation of *Cucumis* species and that IRBs strength correlates positively with genetic distance. These results may also be helpful in establishing a model system that eventually enables the dissection of IRBs genetic control for breeding purposes.

## 2. Results

### 2.1. Patterns of Crossability in Cucumis

IRBs in *Cucumis* were analyzed from a six × six diallel cross experiment using six *Cucumis* species: *C. anguria* L. accession PI282442 (ANG), *C. dipsaceus* (DIP), *C. ficifolius* A. Rich. (FIC), *C. melo* (MEL), *C. pustulatus* Naudin ex Hook.f. (PUS) and *C. zeyheri* (ZEY). Figure 1 represents the patterns of crossability between these species by means of average crossability index (CI) values. CI ranges from fully successful crosses setting fruits with viable seeds (level 1) to crosses impeded by the arrest of pollen tubes at the top of the stigma (level 8). As expected, self-pollen was accepted (reaching the ovaries and fertilizing the ovules) for all six *Cucumis* species producing fertile progenies. The same behavior was found in six additional species (twelve taxa in total, as reported in Appendix A), which supports that SI is absent in the *Cucumis* genus. However, none of the interspecific crosses was fully fertile because of pre- and post-zygotic IRBs acting at different reproductive stages (Figure 1). Hereafter we analyze the role of these IRBs in determining reproductive isolation in *Cucumis*.

### 2.2. Pre-Mating Factors Do Not Correlate with the Average Crossability Index

Pre-mating factors, including the morphology of the reproductive organs, may affect interspecific hybridization success and contribute to reproductive isolation. Accordingly, in this work, we measured four floral traits in the six *Cucumis* species reported above: pollen size and stigma, style, and ovary lengths. Except for style length, all the rest showed high variability among species (Appendix A). According to the statistical analyses (Appendix A; see Section 4), pollen grains from FIC and MEL were significantly bigger (up to 5–25% avg) than those from ANG, DIP, ZEY, and particularly PUS with the smallest grains (Appendix A). Stigma length showed great variability, but MEL stigmas were significantly shorter (1.65 mm avg) than the rest, while ZEY stigmas were clearly the longest ones (2.48 mm avg) (Appendix A). ANG and MEL styles were, on average, much shorter than the rest (up to 45% when compared to PUS) (Appendix A). Lastly, PUS and ZEY had significantly longer ovaries than the rest, while MEL and ANG ovaries were the shortest (Appendix A). As the main result, we found that there was no significant correlation (Pearson’s coefficient not significant at *p* < 0.05) neither between the average CI as male parent and pollen grain size nor between the average CI as female parent, and stigma-style, ovary, or pistil lengths. Therefore, results do not support a clear influence of these floral traits on interspecific crossability.

### 2.3. Pre-Zygotic IRBs in Cucumis Act Predominantly at the Stigma and at the Ovary

A significant part of the interspecific crosses failed because pollen tubes were arrested at different locations along the pistil (Figure 1) but mainly in the stigma and the ovary, as exemplified by Figure 2. More specifically, ‘early’ and ‘late’ arrest sub-types could be distinguished in both tissues. Pollen arrest was not consistently observed in the style, and this might be somehow related to its short length in all tested *Cucumis* species (i.e., 42 to 84% of the stigma length on average). Pre-zygotic compatibility among *Cucumis* species was very variable (Figure 1). For instance, ANG and ZEY were reciprocally compatible at the pre-zygotic level, and DIP was also compatible with these two, but only as a female parent. Meanwhile, FIC was compatible as a female parent with all tested *Cucumis* species, except for MEL, but incompatible with all of them as the male parent. However, PUS showed a certain degree of incompatibility in almost all crosses (except for DIP), and MEL was highly incompatible in most cases. MEL incompatibility was particularly evident when crossed as a female parent since all non-self pollen tubes were arrested early at the MEL stigma (Figure 2A), while MEL pollen tubes were able to reach the ovary in some crosses but without fertilizing ovules in any case (Figure 2C).

### 2.4. Pollen Tube Arrest and Unilateral Cross-Incongruity/Incompatibility

Some pairwise hybridizations exhibit unilateral cross-incongruity/incompatibility (UCI). For instance, DIP pistil accepts ZEY pollen, but DIP pollen is arrested at the ZEY stigma (Figure 3). In contrast, DIP pollen is accepted by PUS (albeit with difficulty) and FIC, but the reciprocal crosses are not viable because PUS and FIC pollen are arrested early in the DIP stigma (Figure 3). UCI is particularly interesting in FIC since FIC pistils accept pollen from DIP, ZEY, ANG, and PUS, while FIC pollen is arrested in all reciprocal crosses.

### 2.5. Post-Zygotic IRBs: Fruit Set, Seed Set, and Seed Viability

Figure 1 and Figure 4 reflect the fruit set percentage in all performed crosses. All self-pollinations (except for FIC) as well as five interspecific crosses (ZEY × ANG, FIC × ANG, FIC × ZEY, FIC × DIP, and PUS × DIP) exceeded the 50% of fruit set. However, a fruit set below 40% was found in five additional interspecific crosses (14% of the total), and no fruit set was detected in the other 20 crosses (56% of the total) (Appendix A). This latter group includes all crosses performed with MEL (as a female or male parent) where pre-zygotic barriers prevent the development of pollen tubes (see Figure 1). Interestingly, in ANG × DIP and PUS × DIP interspecific crosses, we did not observe pollen tubes reaching the ovules 24 h post-pollination (pp), but fruit set percentage reached 40% and 50%, respectively. Thus, ovule fertilization in these two crosses is supposed to occur later on. In contrast, in the DIP × ANG cross pollen tubes were found to fertilize ovules 24 h pp, but the fruit set percentage was very low.

In general, fruits from interspecific crosses produced a significantly lower number of seeds than those from seed parent self-pollinations. Interspecific seeds were also smaller and showed a lower germination rate than seeds from seed parent self-pollinations (Figure 4 and Appendix A). For instance, the seed germination rate was below 10% in DIP × ZEY, FIC × ANG, ZEY × ANG, and below 1% in PUS × DIP (in PUS × DIP, the number of produced seeds was similar to selfed PUS but most of them were empty). Other interspecific crosses produced just a few small seeds that did not germinate at all (e.g., ANG × ZEY) or a few weak seedlings that died quickly (e.g., ANG × DIP and DIP × ANG). However, some interspecific crosses produced a similar number of well-formed seeds than seed parent species, including FIC × ANG, FIC × DIP, FIC × ZEY, and ZEY × ANG, and seeds from FIC × ZEY, FIC × PUS, and FIC × DIP interspecific crosses reached germination rates between 18–30%. (Figure 4 and Appendix A). Clustering of post-zygotic reproductive stages shown in Figure 4 suggests a correlation between germination rate and seed weight as well as among seed number, fruit weight, and fruit set. Additionally, the clustering of interspecific crosses roughly groups those producing F_1_ hybrids (i.e., FIC × ANG, FIC × DIP, FIC × ZEY, ZEY × ANG, and PUS × DIP).

### 2.6. Post-Zygotic IRBs: Fertility in Interspecific Hybrids

A total of eight interspecific F_1_ hybrids were obtained in this work. Seven out of them resulted from the full diallel cross (F × A, F × D, F × P, F × Z, Z × A, P × D, and D × Z) (Figure 1), and another one was obtained by crossing ANG’ (*C. anguria* L. accession CUC27/1983) with DIP (D × A’). Two intraspecific hybrids were also obtained by crossing ANG and ANG’ (A’ × A and A × A’). The interspecific nature of the hybrids was determined by assessing a selected set of diagnostic traits (Figure 5; Appendix A) and further confirmed by PCR-amplifying species-specific SSR markers.

Post-zygotic reproductive barriers are known to include low fertility or sterility exhibited by interspecific F_1_ hybrids. To assess male fertility in F_1_ hybrids, we first estimated the number of pollen grains per flower and found it highly variable but without clear differences between parents and interspecific F_1_ hybrids (Figure 6A). However, pollen viability was uniformly high (>85%) in parents, except for FIC (36%), but dropped to ~10% in the interspecific F_1_ hybrids (Figure 6B,C). Large tetraporate pollen grains (usually associated with high ploidy levels) were observed in FIC and some interspecific F_1_ hybrids (e.g., Z × A) (Figure 6C). All interspecific F_1_ hybrids were male sterile (i.e., did not produce male flowers or pollen viability was rather low), and all FIC hybrids were also female sterile. In contrast, intraspecific A × A’ and A’ × A hybrids derived from crosses between two *C. anguria* accessions previously considered as different varieties [3] were fully fertile (Appendix A).

Female fertility in interspecific F_1_ hybrids was assessed by backcrossing with both parents and by self-pollinating in those hybrids producing male flowers (Z × A, P × D, and F × P). As with the original six × six diallel cross, pre- and post-zygotic barriers (pollen tube growth, fruit set, and seeds viability) were evaluated in backcrosses and self-pollinations of the interspecific F_1_ hybrids. 

Interspecific F_1_ hybrids from UCI crosses also showed UCI in the same direction, i.e., pollinations onto the F1 hybrid only succeeded with pollen from the male parent in the original cross. For example, (Z × A) × ANG backcross is feasible, but (Z × A) × ZEY is not because ZEY pollen tubes are arrested at the Z × A stigma (Figure 7). D × Z and P × D F_1_ hybrids showed similar behavior, though pollen from the hybrid female parent still may produce a few seeded fruits (Figure 7). This trend also becomes apparent in FIC hybrids. Thus, pistil-side UCI in *Cucumis* is suggested to be a dominant trait (e.g., ZEY is dominant over the DIP allele, and therefore heterozygous pistils select against DIP pollen).

To further support these observations, gene action between alleles of the F_1_ hybrid parental species was estimated for different traits (Figure 8). Among diagnostic traits (see also Figure 5), the leaf length/width ratio showed dominance, while others as ovary length showed overdominance, being F_1_ hybrid ovaries longer than those of parents (except for Z × A where the trait is additive). Regarding reproductive traits, pre-zygotic pollen-pistil interaction and fruit set resulting from F_1_ hybrid backcrosses with the seed parent are mostly dominant. For instance, gene action (d/|a|) for pollen-pistil interaction in F × D backcrosses with the seed parent was −0.93, and the means contrast between the F_1_ hybrid and the expected mid-value from FIC and DIP was statistically significant (*Prob* > F 0.014). Thus, FIC pollen was arrested earlier than expected in the F_1_ hybrid, and the DIP allele has a dominant interaction over the FIC allele (Figure 8). In some cases, dominance is apparently partial (e.g., fruit set in (D × Z) × DIP) as the pistil barrier of the homozygote parent (ZZ) is slightly stronger than the heterozygote F_1_ hybrid (DZ), consistent with a gene dosage effect. The Z × A F_1_ hybrid is an exception in which crossability is less successful than both parents, and these two traits are over-dominant (Figure 8).

### 2.7. Pre- and Post-Zygotic IRBs Contribute to Reproductive Isolation (RI) in Cucumis

As previously described, a CI was built in *Cucumis* according to the time of action of the IRBs (Figure 1). In the CI level 1, no IRB is discerned, and the progeny is fully fertile (this level only includes self-pollinations); in levels 2–3, post-zygotic IRBs undermine fruit set, seed germination and interspecific hybrid fertility, and in level 4 seeds do not germinate; lastly in levels 5–6 and 7–8 pre-zygotic IRBs determine the arrest of pollen tubes at the ovary and at the stigma/style, respectively. Thus, IRBs contribute to RI by acting in sequential stages. In this study, a total of 15 species pairs were analyzed for four sequential stages: pre-zygotic pollen-pistil interactions (CI levels 5–8 transformed into a 0–4 scale, see Methods), and post-zygotic fruit set, fruit weight and seed set (CI levels 2–4). In the pollen-pistil interaction stage, pollen tube arrest was fairly common and prevented 20 out of the 30 performed interspecific crosses (including all combinations with MEL). Thus, pre-zygotic isolation mean contribution to RI was relatively high (Table 1). The three post-zygotic isolation stages could only be assessed in eight species pairs since the remaining seven did not set fruit (see Methods). The fruit set isolation stage has the highest contribution to the total post-zygotic RI. Overall, total isolation approaches one, and therefore absolute, and relative contributions are quite similar. Thus, the estimated IRBs are sufficient to cause near-complete RI between most *Cucumis* species.

### 2.8. Genetic Diversity and Phylogenetic Clustering Analysis

Flow cytometry was used to verify the ploidy level of the accessions used in this study. Most *Cucumis* species are described in the literature as diploids, but a few of them have been reported as tetraploids or hexaploids (e.g., *C. ficifolius*, *C. zeyheri*, and *C. pustulatus*) [9]. Ploidy analysis showed that FIC and PRO (*C. prophetarum* L.) accessions were tetraploids, while the rest were diploids (Appendix A). Flow cytometry also suggested consistent differences in genome size among diploid species, but this point should be further confirmed (data not shown).

Filtering of GBS-derived SNP markers provided a set of 10,967 SNP positions for the analysis of diploid parents and hybrids (see Methods). These SNP markers revealed that the heterozygosity level was low in all diploid accessions ranging from ~0.3 to 0.4% except for MEL < 0.1%. As expected, interspecific hybrids showed a significantly higher heterozygosity level when compared with their parents (≥10% in all hybrids except for A’ × A with 1.6%), and this heterozygosity increases with the genetic distance between hybridizing species (e.g., P × D with 13.5% vs. A’ × A with 1.6%) (Appendix A).

FIC and PRO genotypes were identical in >99% of the 10,967 SNPs, and their phenotypes (Appendix A) were also found to be highly similar, suggesting that both accessions may belong to the same species. In addition, both tetraploids show percentages of heterozygosity similar to the hybrids (~12.8%) and present a number of tri-allelic SNPs as detected by GBS. Altogether, these results suggested that FIC and PRO are allotetraploids. Accordingly, hybrids obtained by crossing FIC with diploid accessions were triploids (Appendix A) and showed an even higher level of heterozygosity (ranging from 17.5% in F × D to 19.5% in F × P). This ‘extra’ heterozygosity mainly derives from SNP positions where the fixed allele in all sequenced *Aculeatosi* species (ANG, DIP, PUS, ZEY, and MYR (*C. myriocarpus* Naudin)) differs from the fixed allele in FIC. Heterozygosity also increased with parents’ genetic distance in triploid hybrids.

Nei’s genetic distances among *Cucumis* spp. accessions were estimated by using the same set of 10,967 SNP markers reported above, and a phylogenetic tree was inferred by using the Neighbor-Joining method (Figure 9). A first cluster groups all the species belonging to the *Aculeatosi* section (DIP, MYR, ZEY, ANG, PUS, FIC, and PRO). On closer inspection, three sub-clusters may be appreciated: PUS appears clearly separated from the rest, FIC and PRO cluster together, and the other five species form a rather homogeneous group. Species representative of the other sections also clustered separately, GLO (*C. globosus* C. Jeffrey) and SAG (*C. sagittatus* Wawra and Peyr.) from section *Sagittati*, MAD (*C. maderaspatanus* Siddarthan s.n.) and MEL from section *Cucumis* and MET (*C. metuliferus* E. Mey. ex Naudin) from section *Metuliferi*. 

### 2.9. Correlation between Pre- and Post-Zygotic Isolation and Genetic Distance

This clustering roughly matches the crossability relationships as classified by the CI. To estimate more accurately the relationship between genetic distance and RI, we computed RI values (varying between zero and one) for the four defined pre- and post-zygotic stages in all species pairs (see Section 4). Figure 10A shows plots of correlation between RI values for pollen-pistil interaction and fruit set stages and genetic distance. Pearson’s coefficient supported a significant positive relationship (*p* ˂ 0.01) between both reproductive isolation metrics and genetic distance. Mantel tests also support these correlations, but significance was slightly lower for pollen-pistil interactions (*p* ˂ 0.05) (Figure 10B).

## 3. Discussion

The pattern of crossability observed in the set of *Cucumis* species studied is the result of IRBs acting at pre- and post-zygotic levels throughout the process of flower fertilization. In short, FIC accepts pollen from all tested species except for ME; MEL does not show compatibility with any wild species, and ANG, DIP, ZEY, and PUS are partially cross-compatible among them (see Figure 1). Although with differences, these results are generally consistent with the *Cucumis* crossability polygons determined by den Nijs and Visser et al. [10] and Raamsdonk et al. [11]. Besides that, in this work, we analyzed how these IRBs contribute to RI, their correlation with genetic distance, and the nature of the underlying gene action.

### 3.1. Pre-Zygotic IRBs in Cucumis

Pre-mating factors contributing to RI can be very diverse such as ecogeographic isolation, flowering timing, or differences in floral morphology [24]. The *Cucumis* African wild species studied in this work inhabit two distinct geographic areas separated by the Afrotropical realm and the great Rift valley. Roughly, ANG and ZEY distribution areas overlap in Southeastern Africa, while FIC, PUS, DIP, and MEL (though the MEL accession used here is from Pakistan) overlap in East-Central Africa [3,25]. Unfortunately, the available data on the geographical location of study accessions are not accurate enough to consider this factor for analysis. Morphology traits may include pollen size as well as stigma, style, and ovary lengths [26], whose values were estimated here. Pollen grain size has been related to the pollen provisioning capacity and its potential to cross pistils and reach the ovules. For instance, a strong positive correlation was found between pistil length and pollen volume in the tribe Lycieae (Solanaceae), suggesting that both traits may co-evolve [27]. However, there are frequent exceptions to this tenet, and such a correlation was not found within the tomato clade (*Solanum*) [19]. Neither do we find that correlation in the *Cucumis* species analyzed in this study, but pollen size was found to correlate negatively (*r* = −0.78) with pollen number, as expected to compensate for the resources allocation for the male function [27]. At the interspecific level, FIC and MEL pollen tubes were arrested early in all crosses in spite of deriving from ‘big’ pollen grains. Moreover, even though both species have short stigma-style (and pistil) structures, FIC accepts pollen from all *Aculeatosi* species, while MEL does not accept pollen from any of them. Altogether, results suggest that these floral attributes (pre-mating factors) do not correlate significantly with interspecific crossability in *Cucumis*.

At the post-mating pre-zygotic level, our results show that pollen tube arrest in interspecific incompatible crosses between *Cucumis* species takes place mainly at the stigma or at the ovary and rarely at the style. Within the well-studied genera *Brassica* and *Solanum*, pollen tube arrest in interspecific incompatible interactions also occurs at the stigma [28] or in the upper part of the style [19] respectively. The late-acting arrest of interspecific *Brassica* pollen tubes has also been reported at the ovary level in *Arabidopsis*, apparently because tubes failed to be directed toward the ovules [29]. Incompatible crosses are often characterized by the presence of distorted pollen tubes with swollen tips (e.g., MEL × ZEY, ZEY × FIC, PUS × MEL, ANG × PUS, DIP × PUS, etc.). This phenotype has been associated with ‘active’ pollen tube rejection due to SI or UI (including SC × SC crosses) in *Solanum* [30] and *Nicotiana* [31], but abnormal pollen tube growth could also result from incongruity barriers [32]. 

Under our greenhouse growing conditions, in self- and intraspecific compatible crosses (e.g., ANG × ANG’ and ANG’ × ANG), ovules were usually fertilized in less than 24 h. Thus, the fruit set is mostly impeded in those crosses where pollen tubes did not get to the ovules 24 h pp. Exceptions include ANG × DIP, and PUS × DIP crosses where fertilized ovules were not observed 24 h post-pollination, but the fruit set exceeded 40%, suggesting that DIP pollen growth was retarded in ANG and PUS pistils. However, the weight of the seeds from these two crosses was rather low, and germination mostly failed (ANG × DIP seeds did not germinate, and PUS × DIP seed germination percentage was <1%). ‘Slow growth’ of pollen tubes before reaching the ovules was associated with interspecific incompatibility in *Cucumis* by Kishi and Fujishita [12], but current evidence does not allow to establish a causal link with late-acting pre-zygotic or early-acting post-zygotic barriers in ANG × DIP, and PUS × DIP crosses.

Some *Cucumis* species pairs show unilateral cross-incongruity/incompatibility (UCI) (e.g., DIP × ZEY vs. ZEY × DIP). This phenotype resembles to some extent the unilateral incompatibility (UI) described in the tomato clade (Solanaceae) [27], but it should be considered an exception to the UI rule [33] (i.e., SC × SI functions but SI × SC fails) since all *Cucumis* species are self-compatible (SC). Such exceptions are not infrequent in other species. For instance, in *Solanum*, UI was also reported for SC × SC type crosses between *S. lycopersicum* L. and wild tomato species (*S. habrochaites* S. Knapp. and D.M. Spooner and *S. pennellii* Correll) [30]. UI exemplified by the *S. Lycopersicum* × *S. pennellii* (LA0716) cross is controlled by a pollen rejection mechanism independent of SI and mediated by a farnesyl pyrophosphatase synthase (*FPS2*) [34]. The nature of the *FPS2*-mediated barrier is still unknown, but the authors suggested that it is an active inhibitory mechanism (incompatibility) because the expression is dominant in F_1_ hybrids. Dominant gene action is also exhibited by pistil-side UCI in *Cucumis* F_1_ hybrids, and this might support an active pollen rejection mechanism. However, passive reproductive barriers (incongruity) may explain this phenotype as well [35]. Interspecific incongruity mechanisms have been well-studied in grasses. For instance, in maize, UCI can be defined as a pollen-pistil barrier resulting from incongruity rather than active rejection that unidirectionally prevents hybridization between SC parents. Several UCI systems have been described in maize, but the best known is the *Ga1* (gametophyte factor-1) system that prevents sweet corn varieties pollen (*ga1*) from fertilizing popcorn maize (*Ga1/Ga1*) [36]. Interestingly, a pollen-expressed pectin methylesterase (*PME*) gene at the *Ga1* locus was recently shown to confer the male function [37]. In this system, pollen behavior is gametophytically determined by its genotype (*Ga1* or *ga1*), while the pistil barrier is sporophytically determined in heterozygous plants. Therefore, hybrid plants (*Ga1/ga1*) are semi-compatible in backcrosses to popcorn, and *ga1* pollen is said to be incongruous on *Ga1/*—pistils [38]. The UCI characterized in *Cucumis* may also be consistent with this model since seed parent pollen fails to grow in F_1_ hybrid pistils (i.e., pistil-side UCI is dominant).

### 3.2. Post-Zygotic IRBs in Cucumis

The effects of post-zygotic IRBs in the reproductive isolation and speciation processes have been reported in numerous species. For instance, seed and pollen infertility is known to significantly reduce hybrid fitness and contribute to interspecific reproductive isolation in *Solanum* [39]. In *Cucumis*, interspecific post-zygotic barriers lead firstly to reduced fruit set and to a lower number (and average weight) of seeds per fruit when compared with self-pollinations. Subsequently, the viability of the produced seeds is frequently compromised (underdeveloped or empty seeds). Lack of well-formed seeds could still be the result of pre-zygotic late-acting barriers hindering ovule effective fertilization by pollen tubes, but early post-zygotic IRBs impairing post-fertilization development of the embryo and/or the endosperm may also be present. Furthermore, germination success is often jeopardized in hybrid seeds that generally show lower performance. Lastly, interspecific hybrids, when obtained, are male and/or female sterile as observed in previous studies [9,10,13]. In contrast, intraspecific hybrids derived from crosses between two ANG accessions (A’ × A and A × A’) are fully fertile in agreement with a positive correlation between RI and genetic distance (discussed below).

Thus, post-zygotic barriers in *Cucumis* include embryo mortality, hybrid inviability (e.g., A × D and D × A hybrid seedlings dying after the transplant), and sterility. Additionally, most F_1_ hybrids derived from species pairs showing UCI exhibit asymmetric backcross incompatibility. Dominant gene action is suggested for F_1_ hybrid backcrosses with the seed parent assessed at pre- (pollen-pistil compatibility) and post-zygotic (fruit set) stages. Evidence is provided by all three flowering and male-sterile interspecific hybrids Z × A, D × Z, and P × D that fully accept pollen from the male parent but present diverse IRBs to the female parent. *Cucumis* species pairwise showing UCI are genetically close (all of them belong to the *Aculeatosi* section) [10], and basic physiological requirements for pollen tube growth between two species showing UCI are expected to be similar [40]. Altogether, this tempts us to speculate that UCI in *Cucumis* might be under an oligogenic control which would constitute an accessible system for dissecting the genetic control of IRBs.

### 3.3. Reproductive Isolation and Genetic Distance

We found that both post-mating pre- and post-zygotic IRBs (acting sequentially from pollen-pistil interaction to fruit and seed set processes) contribute to reproductive isolation among *Cucumis* species. Isolation expressed prior to fertilization was stronger, but post-zygotic mechanisms were also found to contribute significantly to reproductive isolation. Due to the sequential action of post-zygotic barriers, the first post-zygotic acting factor (fruit set capacity) showed the highest influence on isolation when compared with the subsequent stages (fruit size and seed set). In agreement with other studies [23,41], reproductive barriers acting before F_1_ hybrid formation in *Cucumis* prevent most gene flow (>99%), but F_1_ hybrids still exhibit low pollen fertility. However, in stark contrast with that observed in *Nolana*, where pre-zygotic isolation was weak [41], pollen-pistil stage contribution to RI in *Cucumis* exceeded that of all post-zygotic stages. Some other pre-zygotic factors not covered in this study (including ecogeographic and temporal isolation) may also affect the probability of interspecific crossing in *Cucumis*, but barriers at pollen-pistil interaction and fruit set stages seem to be determinant. In addition, greenhouse growing conditions may somehow affect the reproductive behavior of *Cucumis* wild species, and this could be considered another potential constraint of our study. To our knowledge, natural *Cucumis* hybrids have not even been reported between species-sharing distribution areas (e.g., ZEY and ANG) [3]. This suggests that joint IRB contribution maintains species isolation and identity. Nonetheless, a few *Cucumis* species are reported to be polyploids (including *C. zeyheri*, *C. ficifolius*, *C. pustulatus*, etc.) [9] that could result from interspecific hybridization followed by whole-genome duplication [42] as suggested for the allotetraploid *C. ficifolius* studied in this work.

Our results support that strength of RI stages correlates positively with genetic distance in *Cucumis* as reported in only a few other genera [23,41,43]. The *Cucumis* phylogeny inferred from a clustering analysis based on GBS-derived SNP markers is basically consistent with *Cucumis* phylogenies previously obtained with plastid and nuclear ribosomal markers [2,19,22]. Representatives of the different sections roughly cluster as expected. However, the distribution within the *Aculeatosi* section differs partly from previous works, which may be due to the use of different accessions and marker types. In particular, FIC is fairly close to all *Aculeatosi* members and, at the same time, significantly closer to the other sections than the rest. This, along with its heterozygous allotetraploid nature, may suggest that FIC originated from an interspecific cross between two distantly related *Aculeatosi* species or between one *Aculeatosi* species and another one from a different section. Much attention was focused on FIC because of its reproductive behavior. In this respect, the tetraploid nature of FIC could be speculated to contribute to pistil receptivity in interspecific crosses, as suggested by Kho et al. [4]. However, though polyploids are known to self-fertilize more than diploids in general [44], no clear evidence supports a relationship between polyploidy and interspecific compatibility. For instance, the interspecific crossability of diploid ZEY accessions did not seem to differ significantly from that of tetraploid ZEY [10]. Thus, further research will be needed to discern if the allotetraploidy condition is involved in interspecific compatibility in FIC.

Overall, it can be concluded that RI in *Cucumis* results from a network of IRBs acting sequentially at pre- and post-zygotic levels that may include incompatibility and incongruity overlapping processes. The strength of these IRBs correlates with genetic distance between species, and UCI is observed between close species pairs. Exceptionally, the allotetraploid *C. ficifolius* exhibits UCI in all four interspecific combinations with *Aculaetosi* species following the same pattern: it accepts pollen from all these species while its pollen is not accepted by any of them. In-depth characterization of the different UCI systems may facilitate genetic analysis of IRBs in *Cucumis*, shedding light on the speciation process and favoring gene introgression from wild germplasm into cultivated forms.

## 4. Materials and Methods

### 4.1. Plant Material

Thirteen *Cucumis* taxa (including two *C. anguria* accessions) and ten *Cucumis* interspecific hybrids were used in this study (Appendix A). According to Endl et al. [22], four sections of the *Cucumis* genus were represented: *Aculeatosi* (8 spp), *Cucumis* (2 spp.), *Sagittati* (2 spp.) and *Metuliferi* (1 spp.). Five African wild *Cucumis* species from the *Aculeatosi* section (*C. anguria*, *C. dipsaceus*, *C. ficifolius*, *C. zeyheri*, and *C. pustulatus*) were selected for phenotyping the IRBs because of their relative phylogenetic proximity and their diverse crossability relationships [10,19,20]. The wild *C. melo* accession Ames 24,294 from North Central Regional Plant Introduction Station (Ames, IA, USA) was chosen for phenotyping IRBs to avoid possible effects of domestication or breeding on IRBs displayed by cultivated accessions. Interspecific hybrids were obtained in this work (Appendix A).

### 4.2. Plant Cultivation and Handling

Before sowing, forceps were used to break open the seed coat, and then seeds were sterilized by immersion in NaClO 1% (*w*/*v*) solution for 1 min. Afterward, seeds were rinsed twice with distilled water and placed in Petri plates with wet cotton and filter paper. Seeds were kept at 37 °C in darkness conditions for 24–48 h to enhance germination. Seedlings were transplanted into pots at the cotyledon stage and placed in a growth chamber before moving them to the greenhouse 2–3 weeks later. Five plants of each taxon (including F_1_ interspecific hybrids) were drip-irrigated in peat pots and grown under standard glasshouse conditions at the COMAV-Universitat Politècnica de València (València, Spain) facilities during the spring and summer (April–August) in 2017, 2018, and 2019. All phenotypic data were recorded over this period.

Artificial pollinations were performed using male and female flowers that were bagged 24 h before anthesis to prevent non-controlled pollinations. Pollinated female flowers were subsequently re-bagged for 24 h post-pollination (pp). Direct and reciprocal crosses were made among the six selected *Cucumis* accessions, and selfings were used as positive controls. All interspecific crosses (30) and selfings (6) were performed at least 5 times to evaluate pre-zygotic IRBs. Pollinated pistils were collected at 24 h pp (just occasionally 48 h pp) since we previously found pollen tubes reaching the ovules mostly before 24 h in selfings and compatible crosses under our growing conditions. To evaluate post-zygotic IRBs, all single crosses and selfings were performed at least 25 times (except for 5 crosses with ≥17 times) and up to 46 times. The number (fruit set) and weight of fruits, as well as the number (seed set) and weight of seeds per obtained fruit, were recorded for each single cross. When produced, seeds were sown to estimate germination percentage. Germinated hybrid plantlets were grown under greenhouse conditions to evaluate their reproductive behavior: the ability to produce male and/or female flowers and ability to set fruit with viable seeds when selfed and backcrossed with both parents. When possible, hybrid backcrosses and selfings were performed at least 5–10 times.

### 4.3. Reproductive Isolation Measures

The contribution of IRBs to reproductive isolation between *Cucumis* species was assessed at one post-mating pre-zygotic stage (pollen–pistil compatibility) and three post-zygotic stages (fruit set, fruit weight, and seed production). These four stages can be considered sequentially acting stages in the life history where an IRB only can prevent gene flow that was not already eliminated by previous stages of isolation [23]. The degree of compatibility of interspecific crosses for each isolation stage was estimated relative to self-pollinations (intraspecific controls) according to the equation described in Jewell et al. [41]: Reproductive Isolation (RI) = 1 – (average success of interspecific crosses/average success of self-pollinations). Therefore, RI indices vary between 0 (interspecific cross is as compatible as self-pollination) and 1 (completely incompatible interspecific crosses). Additionally, as described by Jewell et al. [41], when interspecific crosses were more successful than self-pollinations (just a few cases), the resulting negative RI values were set to 0. Direct and reciprocal crosses between every pair of species were averaged to give a mean isolation score for each isolation stage.

Pre-zygotic pollen-pistil compatibility was evaluated by observing pollen tube growth in pistils collected 24 h pp under fluorescence microscopy and recorded on a five point-scale (Appendix A): 0 = pollen tubes halted before the stigma half; 1 = pollen tubes halted at the stigma end or the style; 2 = pollen tubes reaching the distal part of the ovary; 3 = pollen tubes reaching the proximal end of the ovary; 4 = pollen tubes fertilizing the ovules. Pollen-pistil compatibility index value for each cross was estimated using the mean of all scores standardized by the mean of self-pollen performance in the seed parent pistils. Post-zygotic compatibility was evaluated at three stages (Appendix A): fruit set (percentage of developing fruits obtained for each cross), fruit weight (measured in gr), and seed set (number of formed seeds per mature fruit).

Absolute (AC) and relative contribution (RC) of a RI component at stage n in the life history and total reproductive isolation (T) for m RI components were calculated using the equations described in Ramsey et al. [23]: AC_*n*_ = RI = RI_*n*_ (1 − $\overset{n-1}{\underset{i=1}{\sum }}\mathrm{ACi}C_{i}$) and T = $\overset{m}{\underset{i=1}{\sum }}AC_{i}$ for *m* sequentially acting components. Relative contributions were calculated as RC_*n*_ = AC_*n*_/T.

### 4.4. Fluorescence Microscopy

Flowers were collected 24–48 h pp. The ovary aculei were removed from collected flowers, and a ~1 mm layer section of the gynoecium covering the stigma, style, and full ovary was dissected. Each section was fixed in a 3:1 ethanol (96%): acetic acid solution for 24 h and subsequently stored in ethanol at 70% for the long term. Samples were rehydrated (by rinsing in ethanol 50%, 30%, and distilled water) and treated with an alkaline solution (NaOH 8M) for 24 h and afterward washed in distilled water for 10 min. Gynoecia were then stained for 24 h in darkness using an aniline blue solution (1:10 *w*/*v* aniline blue in 50 mM K_2_HPO_4_/KH_2_PO_4_ buffer, pH 7.5). Each gynoecium section was squashed for observation of pollen tubes with a LEICA DM5000 fluorescence microscope (Leica, Munich, Germany). Single images were mostly obtained at 25× magnification with LAS V4.9.0, and pistil composite images were generated by splicing individual frames together using Adobe Photoshop, version CS5 (Adobe Systems Inc. San Jose, CA, USA). ImageJ softwa re [45] was used to measure the length of stigmas and styles (at least 10 per accession). Pollen grain size (~200 grains per accession) and pollen viability were estimated after lactophenol blue staining. At least three flowers (biological replicates) were sampled from each accession. Anthers were dissected with forceps and placed into Eppendorf tubes that were vortexed for 10 s and spun for 5 s. Then, 50 µL of lactophenol blue staining solution [46] containing 200 mg/mL aniline blue in 1:1:1:1 (phenol: lactic acid: glycerol: distilled water) solvent was added to each tube. After 30 min at room temperature, tubes were vortexed for 10 s and spun for 5 s again. Anther tissue was then removed, and four aliquotes (technical replicates) of 2.5 µL were sampled from each tube. Samples were observed under bright field in a LEICA DM5000 microscope. Stained pollen grains were recorded as viable, and small, distorted, and unstained pollen grains as non-viable. The pollen grain size was also measured by using ImageJ software [45].

### 4.5. DNA Extraction, Ploidy Detection, and SSR Analyses

Fresh leaf samples were collected from all the *Cucumis* taxa (named as in [47]) and interspecific hybrids used in this work (Appendix A). DNA was extracted following a modified CTAB method [48]. 

The ploidy level of all accessions was determined by flow cytometry in a ParteCyflow ploidy analyzer (Partec GmbH, Münster, Germany) using Partec CyStain based on DAPI (4′-6-diamidino-2-phenylindole) staining of the DNA. Approximately 0.5 cm^2^ of young leaf tissue were finely chopped with a sharp razor blade in a plastic petridish and then 400 µL of nuclei extraction buffer (Partec CyStain UV precise P) were added. After a short incubation (30 s to 5 min) extracts were filtered through a 40 µm mesh filter, mixed with 1.6 mL of staining buffer (Partec CyStain UV precise P) and kept on ice. Flow cytometry was performed with a medium flow rate (0.4 µL/s) and samples were excited with UV light and analyzed in the blue parameter according to the manufacturer’s protocol. Fluorescence of well known diploid accessions was used as reference to assess ploidy levels of the rest (*C. dipsaceus* PI236468 and *C. melo* Ames24297/TRI 5-4).

A set of four SSR markers (CMCT168, CMGA172, CMACC146 and CMAGN39) developed from melon [49] were PCR-amplified and used to specifically distinguish *Cucumis* species and hybrids. SSR amplifications were performed in a Techne TC-512 thermal cycler (Techne Inc. Burlington, NJ, USA) in a total volume of 25 µL of reaction buffer containing 1× Netzyme DNA polymerase reaction buffer, 2 mM MgCl_2_, 200 µM of each dNTP, 500 nM of the reverse and the fluorophore labelled M13 primers and 125 nM of the M13-tailed forward primer, 1.9 U Netzyme DNA polymerase (Biotools, Madrid, Spain) and ~20–50 ng of genomic DNA. PCRs were performed as described by Schuelke [50]. Cycling conditions were as follows: an initial denaturation step at 94 °C for 5 min followed by 30 cycles at 94 °C 30 s, 56 °C 45 s and 72 °C 45 s and then 8 cycles at 94 °C 30 s, 53 °C 45 s and 72 °C 45 s with a final cycle at 72 °C for 10 min. Allele lengths were determined using an ABI Prism 3130 Genetic Analyzer with the aid of GeneMapper software, version 4.0 (Applied Biosystems, Foster city, CA, USA).

### 4.6. Genotyping-by-Sequencing

DNA aliquots from all the *Cucumis* taxa and F_1_ hybrids (except for A × A’) (Appendix A) were sent to LGC Genomics Gmbh (Berlin, Germany) for GBS analysis. Two-three technical replicates from each sample were sequenced except for FIC hybrids, D × A’, A’ × A, Z × A, ZEY, MYR, and GLO with only one. ApeKI methylation-sensitive restriction enzyme was used for library preparation. Illumina sequencing adaptors (P5 and P7) and sample-specific barcodes were then ligated to the resulting fragments’ sticky ends. Several rounds of Polymerase chain reactions were performed for library construction. Illumina NextSeq 500 V2 (Illumina, Inc., San Diego, CA, USA) was used for library sequencing, and 75 bp simple reads were generated. A total of nearly 3 million raw sequences were obtained per sample. Sequencing raw data are available in the NCBI SRA (Sequence Read Archive) under Bioproject ID PRJNA849990 (accession numbers SAMN29152053-88 for the 36 *Cucumis* spp. Sequenced samples). SNP calling was carried out in the Bioinformatics service of Instituto de Biología Molecular y Celular de Plantas (UPV, Valencia, Spain). GBS-filtered reads were aligned against the melon reference genome CM3.6.1. (www.melonomics.net accessed on 21 November 2022) using the program BWA mapper [51]. SNP individual variation was detected using Freebayes [52]. Low-quality SNPs were filtered out by a minimum number of reads per SNP (>3) and minimum mapping quality per SNP (>20). The obtained SNPs were converted into Variant Call Format file (VCF). A total of 236,775 identified SNPs among the 36 sequenced samples were visualized using the software TASSEL (Trait Analysis by aSSociation, Evolution and Linkage) [53].

### 4.7. Phylogenetic Analysis

Phylogenetic analyses were performed using GBS-derived SNPs filtered by TASSEL. Sample technical replicates were merged together, and then variants were sequentially filtered. First, a minimum genotyped depth of 10 was selected, and all positions containing missing data or showing no variants were removed. Then, the heterozygosity level was restricted to 0.80 (to reduce mapping bias), and the TASSEL ‘thin sites by position’ option was used to thin out SNPs with a 20 bp minimum distance across the whole genome. Following this procedure, an SNP set containing 10,967 SNPs was generated for the 12 *Cucumis* taxa used in this study (Appendix A), where FIC and PRO tetraploid SNP genotypes were scored as pseudodiploids (i.e., all heterozygous classes were scored as AB). Subsequently, this set was used to calculate the Nei’s [54] genetic distances among *Cucumis* species by means of “VCFR” [55] and “adegenet” [56] packages in R [57]. The additive phylogenetic tree was inferred using the Neighbor Joining (NJ) method with 1000 bootstrap replicates by means of the PoppR [58] package in R.

### 4.8. Statistical Analyses

General statistical analyses were performed using the packages “ggplot2” [59] and “dplyr” [60] in R. Values recorded for pre-mating factors (Appendix A) and post-zygotic IRB parameters (Appendix A) were subjected to one-way analysis of variance (ANOVA) to assess significant differences among their means. The Levene’s test and the Shapiro-Wilk test were used to check equal variances (*p* > 0.05) of data and normal distributions (*p* > 0.05) of residuals of the ANOVA model, respectively. When significant differences among means (*p* < 0.05) were found, Tukey’s multiple range test was carried out. Then, if the assumption of homogeneity of variances could not be assumed, the Welch test was carried out. In the cases where ANOVA residuals were not normal, the nonparametric Kruskal-Wallis test was used to check significant differences among data instead of ANOVA (Appendix A). When neither the requirement of homoscedasticity nor normality was fulfilled, Bonferroni-corrected pair-wise Mann-Whitney-Wilcoxon tests were carried out to detect significant differences (Appendix A). Relationships between reproductive isolation and genetic distance were evaluated by performing Mantel tests with R [57] and by estimating Pearson򲀙s correlation coefficient (*r*) and its *p*-value with Excel (Microsoft, Redmond, WA, USA).

Statistical analyses for studying gene action were carried out with JMP^®^ Version 11 (SAS Institute Inc. Cary, NC, USA). The gene action (degree of dominance) was calculated as the ratio d/|a| where d is the deviation between the phenotype of the F_1_ hybrid and the mean phenotype of the parents (d = F_1_ − [P_1_ + P_2_]/2) and |a| is the absolute additive value (|a| = abs |P_1_ − P_2_|/2). One-way ANOVA was used to detect significant differences among trait Least Squares means for the three genotypes (F_1_, P_1_, and P_2_) and the Student’s *t*-test to determine significant differences between LS means. The contrast F_1_ − 1/2P_1_ − 1/2P_2_ = 0 was used to determine the statistical significance of the gene action by calculating an F-ratio statistic.

A hierarchical clustering heatmap representing post-zygotic IRBs in interspecific crosses (rows) was obtained on phenotypic data (columns: fruit set, fruit weight, seed number, seed weight, and germination rate) by using the ClustVis web tool [61]. Heatmap rows and columns were clustered using correlation distance and average linkage. Unit variance scaling was applied to columns.

## Figures and Tables

**Figure 1 plants-12-00926-f001:**
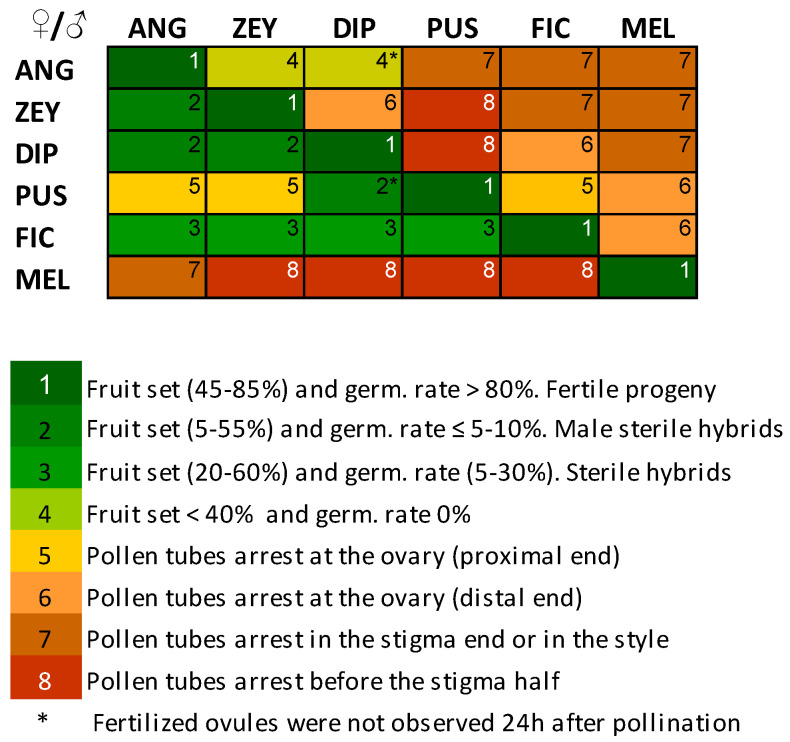
Patterns of crossability in a complete 6 × 6 diallel cross between *Cucumis* species, including both reciprocal crosses and self-pollinations (table’s diagonal). Crossability index (CI) is indicated by numbers (1–8) and color grading (green to red) and ranges from crosses showing no IRBs (level 1) to crosses impeded early after pollination (level 8). CI is determined by pre-zygotic IRBs that result in pollen tube arrest at different pistil parts (5–8) and post-zygotic IRBs that condition fruit set success, seed viability (germination rate), and hybrid fertility (1–4). Fertilized ovules were observed 24 h after pollination in all crosses with CI 1–4 (green color grade) except for those labeled with *.

**Figure 2 plants-12-00926-f002:**
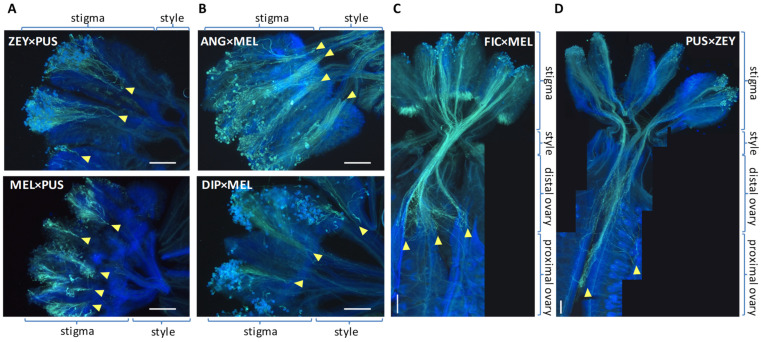
Pollen tube arrest in *Cucumis* interspecific crosses 24 h post-pollination. (**A**) Pollen tubes arrest before the stigma half, (**B**) at the late stigma or the style, (**C**) at the distal ovary, and (**D**) at the proximal ovary. Pictures correspond to the following crosses: (**A**) ZEY × PUS and MEL × PUS, (**B**) ANG × MEL and DIP × MEL, (**C**) FIC × MEL and (**D**) PUS × ZEY. Yellow arrowheads indicate sites of pollen tube arrest. Scale bars correspond to 500 µm.

**Figure 3 plants-12-00926-f003:**
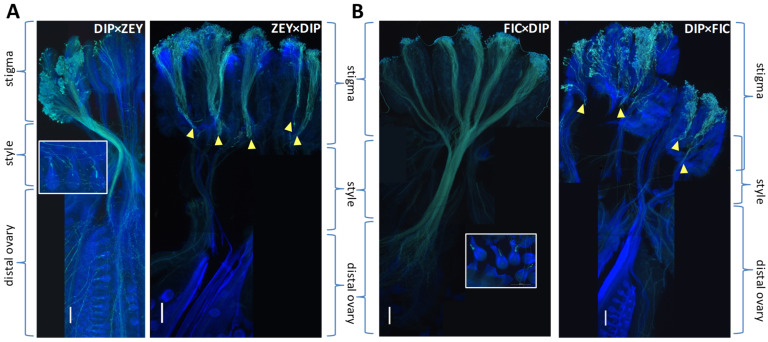
Unilateral cross-incongruity/incompatibility in *Cucumis*. (**A**) Hybridization between DIP and ZEY. Direct DIP × ZEY and reciprocal ZEY × DIP crosses are shown. (**B**) Hybridization between FIC and DIP. Direct FIC × DIP and reciprocal DIP × FIC crosses are shown. Insets show details of fertilized ovules in the DIP × ZEY, and FIC × DIP crosses. Yellow arrowheads indicate sites of pollen tube arrest. Scale bars correspond to 500 µm.

**Figure 4 plants-12-00926-f004:**
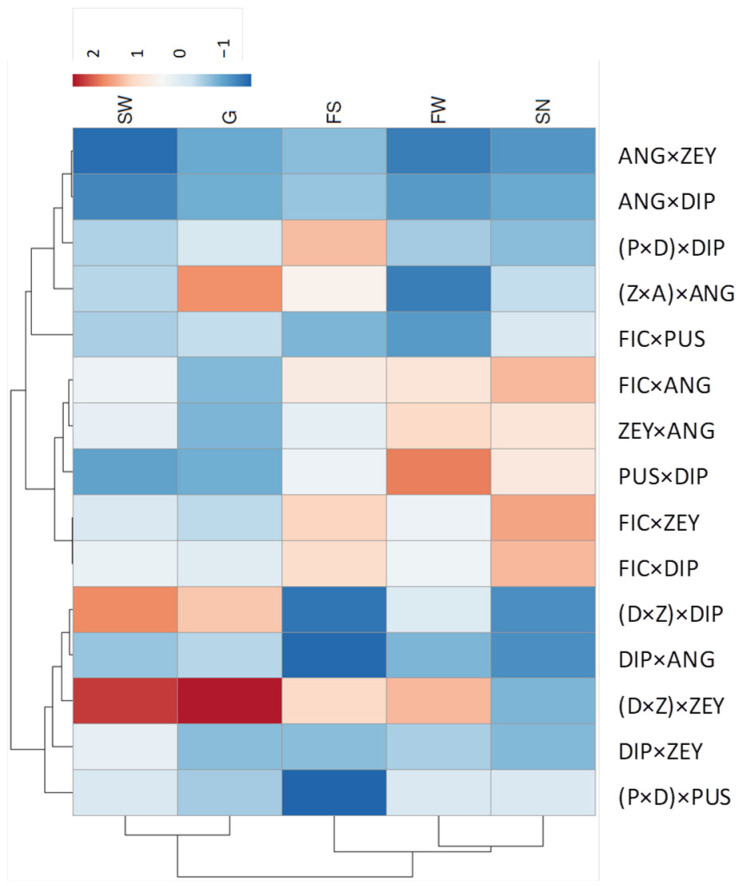
Heatmap of post-zygotic IRBs acting at different reproductive stages in interspecific crosses. Estimated parameters (columns) include seed weight (SW), germination rate (G), fruit set % (FS), fruit weight (FW), and seed number (SN). Interspecific crosses (rows) include those producing at least one fruit. Values for each parameter were estimated relative to seed parent intraspecific controls. The color range from red to blue represents weaker to stronger IRBs to hybridization.

**Figure 5 plants-12-00926-f005:**
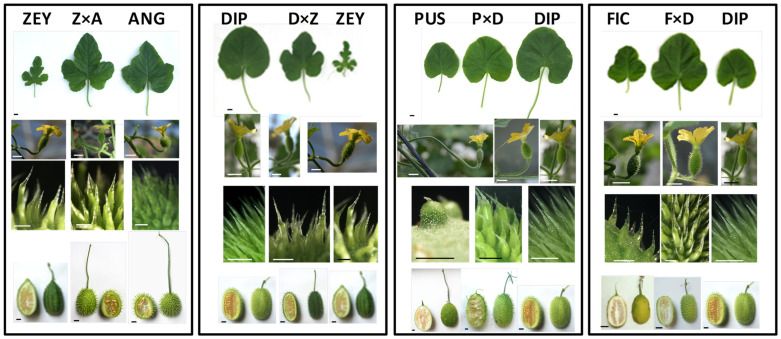
*Cucumis* interspecific hybrids. Picture panels illustrate four diagnostic traits (from **top** to **bottom**: leaf shape/stem length; female flower; ovary aculei (opaque/hyaline portions); and fruit shape/pedicel length) in the diploid interspecific hybrids Z × A (*C. zeyheri* × *C. anguria*), D × Z (*C. dipsaceus* × *C. zeyheri*) and P × D (*C. pustulatus* × *C. dipsaceus*) and the triploid hybrid F × D (*C. ficifolius* × *C. dipsaceus*) and in their parents. Scale bars on the bottom left corner correspond to 1 cm, except for ovary aculei corresponding to 1 mm.

**Figure 6 plants-12-00926-f006:**
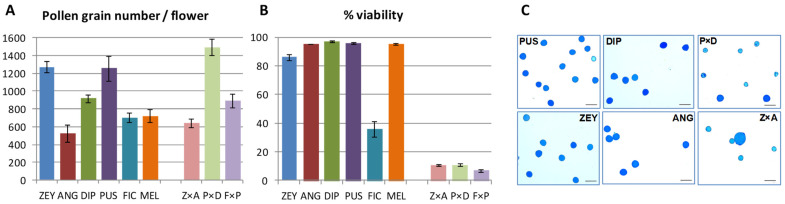
Pollen fertility in *Cucumis* spp. accessions and the interspecific hybrids producing male flowers. (**A**) Total pollen grains per flower. (**B**) Pollen viability estimated as the ratio [fertile pollen grains: total pollen grains]. Bars represent the average and standard deviation estimated from three male flowers for each accession. (**C**) Lactophenol blue staining of fertile (dark blue) and non-fertile pollen (soft blue) grains from the interspecific hybrids P × D and Z × A and their parents. Scale bars on the bottom right corner correspond to 100 µm.

**Figure 7 plants-12-00926-f007:**
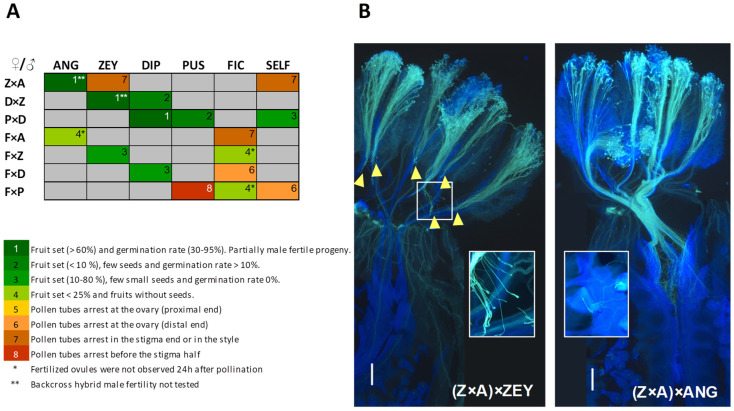
Backcrossing success of *Cucumis* interspecific hybrids. (**A**) Crossability index (CI) is indicated by numbers (1–8) and color grading (green to red) and ranges from crosses showing no IRB (level 1) to crosses impeded early after pollination (level 8). CI is determined by post-zygotic IRB that condition fruit set success, seeds viability and hybrid fertility (1–4) and pre-zygotic IRB that result in pollen tube arrest at different pistil levels (5–8). SELF column shows results of self-pollinations, when possible, and grey cells indicate not performed crosses. Fertilized ovules were observed 24 h after-pollination in all crosses with CI 1–4 (green color grade) except for those labeled with *. (**B**) Pollen tube growth in backcrosses between the hybrid Z × A and their two parents: (Z × A) × ANG and (Z × A) × ZEY. Insets show details of distorted and swollen pollen tubes in (Z × A) × ZEY and fertilized ovules in (Z × A) × ANG backcrosses, respectively. Yellow arrowheads indicate sites of pollen tube arrest. Scale bars correspond to 500 µm.

**Figure 8 plants-12-00926-f008:**
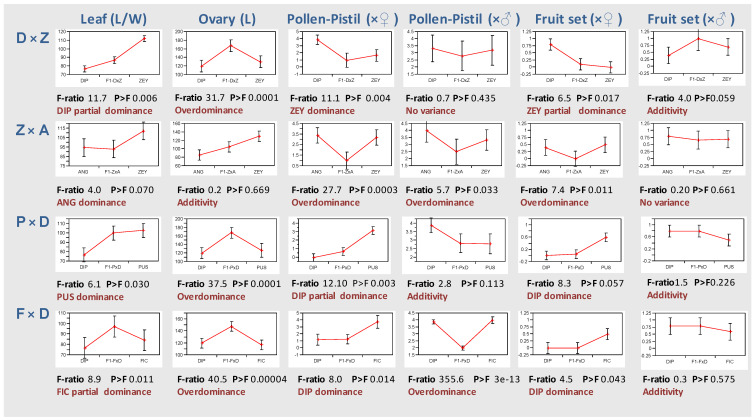
Gene action in interspecific F_1_ hybrids. Least Squares Means for different traits were estimated to compare F_1_ hybrids with their parents. LS Means plots for 4 F_1_ hybrids (rows) and 6 traits (columns) are shown. Traits include leaf length/width ratio (%) and ovary length (mm), as well as pre-zygotic pollen-pistil interaction (CI levels 5–8 transformed into a 0–4 scale, see Section 4) and fruit set (expressed as decimals) in backcrosses with seed parent (×♀) and male parent (×♂). Each plot represents LS means and their standard errors for the parents and the resulting F_1_ hybrid. F-ratios and corresponding *p*-values for the means contrasts between the F_1_ hybrids and the expected mid-value from parents are also indicated.

**Figure 9 plants-12-00926-f009:**
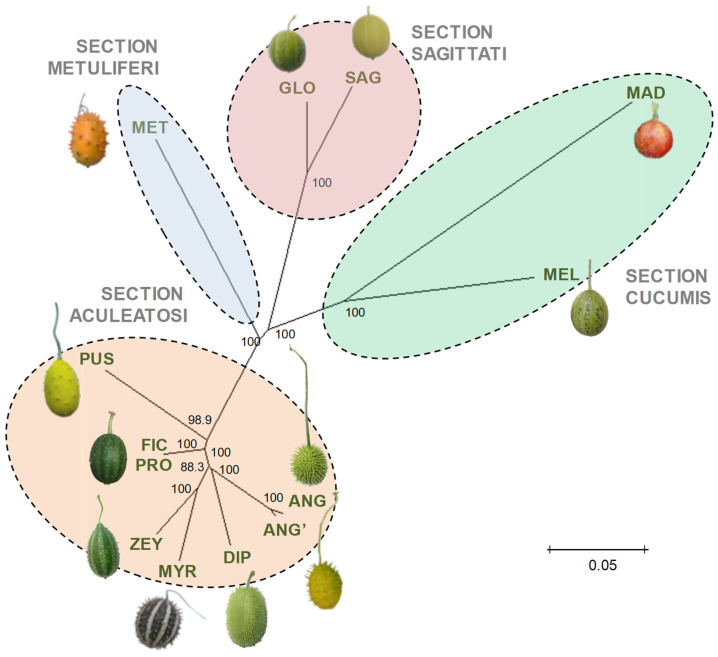
NJ phylogenetic tree based on the genetic distances among *Cucumis* spp. accessions estimated from GBS data. Bootstrap values with 1000 replicates expressed as a percentage (%) are shown at the nodal branches.

**Figure 10 plants-12-00926-f010:**
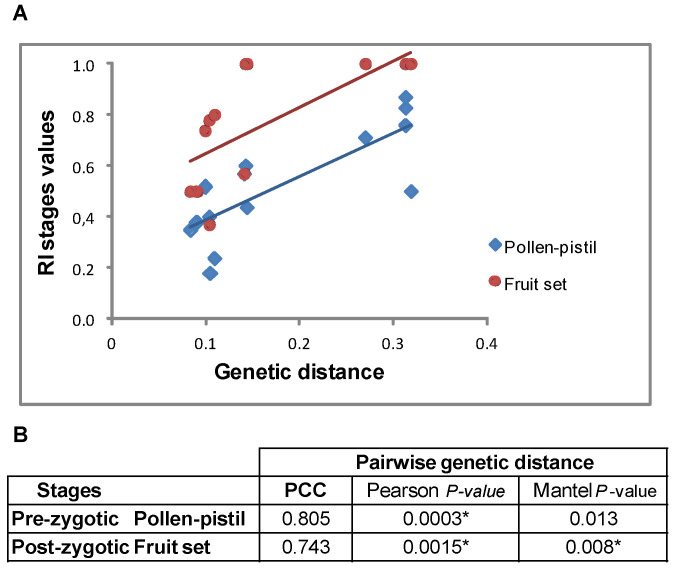
Relationship between pairwise genetic distance and reproductive isolation stages. (**A**) Correlation plots between Nei’s (1972) genetic distance and RI values for pollen-pistil interaction and fruit set stages; (**B**) Statistical significance of the relationship between pairwise genetic distance and RI stages assessed by Pearson’s correlation coefficient (PCC) and Mantel tests (significant values *p* < 0.01 *).

**Table 1 plants-12-00926-t001:** Absolute and relative mean contribution to RI of sequential pre- and post-zygotic isolation stages in *Cucumis*.

			Mean Contribution ^a^	
Sequencial Stages		N	Absolute	Relative
Pre-zygotic	Pollen-pistil	15	0.515	0.519
Post-zygotic	Fruit set	8	0.289	0.291
	Fruit weigth	8	0.110	0.111
	Seed viability	8	0.078	0.079
	Total Post-zygotic	8	0.477	0.481
Total RI			0.992	1.000

^a^ Absolute and relative mean contribution to RI were calculated according to Ramsey et al. [23] (see Section 4).

## Data Availability

Sequencing raw data are available in the NCBI SRA (Sequence Read Archive) under Bioproject ID PRJNA849990 (accession numbers SAMN29152053-88 for the 36 *Cucumis* spp. Sequenced samples).

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
