# Peer review of "Pre- and Post-Zygotic Barriers Contribute to Reproductive Isolation and Correlate with Genetic Distance in Cucumis"

_plants, 2023, doi:10.3390/plants12040926_

Round 1

Reviewer 1 Report

Understanding interspecific reproductive barriers (IRBs) is a key concept in evolutionary biology and can also be informative for improving breeding. However, we currently know very little about which reproductive stage is driving reproductive isolation. The study does a great job at describing the pre- and post-zygotic IRBs present in Cucumis African wild species to give us an idea on the mechanisms that might be underlying speciation.

This is an exciting study, I have mostly minor suggestions for improvement of the manuscript. My main suggestion would be to integrate the significance of this research in the fields of evolutionary biology/plant breeding more throughout the manuscript. You touch on it in the final sentence of your abstract and the end of your discussion but I think your manuscript would improve from further development of the idea in your abstract, introduction and discussion.

Specific comments

Line 14 - Gap in knowledge needs to be made clear in the first couple of sentences of abstract.

Line 89 – what is n referring to here?

Line 103 – which works don’t support Kirkbride? You initially said that paper on Line 94 supported Kirkbride.

Line 133 – “(5-8)”

Line 134 – “(1-4)”

Line 134 – To improve clarity, add to Figure legend “Fertilized ovules were observed 24h after pollination” and change * to “Fertilized ovules were observed 48h after pollination”

Line 144 – FIC is not significantly shorter based on the figure.

Line 146 – From the figure only MEL is significantly shorter. Please make it clear what is significantly different in this results section. Where are the ANOVA results for each comparison?

Line 149 – Why is it roughly?

Line 152 – These figures don’t look at the association between these variables so I wouldn’t refer to them here.

Line 170 – Fig. 2 – I found in Fig. 3 where you show where the sigma, style and ovary is very helpful to orientate myself, can you include this in Fig. 2 as well?

Line 174 – Add what the yellow arrow heads are in the figure legend

Line 190 – Fig. 4 – Why have the names changed for the species? What are they now? It would be easier to follow the paper if the names were kept the same throughout.

Line 192 – You need to refer to Fig. S2 here for fruit set %

Line 197 – Have you put the * in the wrong spot in Fig. 1? This line doesn’t match the figure currently.

Line 198 – What do you mean by significant here?

Line 205 – Fig. S2 - I think it would be clearer to have germination % in a different figure and to change species names in this figure to keep consistent with text (and most of paper).

Line 213 – Explain how clustering was done here.

Line 215 – on one side of what, explain to increase clarity.

Line 220 - Fig. 4 figure legend – explain parameters in order in which they appear in the figure. Also, explain what a value of 2 (red) and -1 (blue) mean and how the phylogeny was created.

Line 234 – Fig. 5 figure legend. Add the fact that ‘parents’ and interspecific crosses are shown. Why is DIP parental stem length in FxD cross so much smaller than the other two?

Line 252 - Fig 6. If you could order the bars by the order in which you have the hybrids it would be easier to find the two parents to compare the height of the bars.

Line 257 – Is PUS not one of the parents? Fix the writing here to increase clarity.

Line 251 – where are these results?

Line 272 - Figure 7 – Keep CI index in same order as Figure 1 e.g. 1 to 8.

313-314 – what do you mean by this sentence, clarify

Line 326 - Table 1 – Add to table legend, how absolute and relative mean contribution to RI were calculated.

Line 332 – Was ploidy in Table S1 based on flow cytometry? Add this to table legend.

Line 337-341 – where is the results table for this?

Line 354 – Briefly how is Nei’s genetic distance calculated?

Line 383 - Figure 10 legend – you mean p>0.01? Usually done the other way round where significant values get *

Line 410 - What do these results mean in a broader context?

Line 488-489 – why oligogenic control?

Line 501 – what stage was more common for Nolana?

Line 563 – instead of glass greenhouse is it just a glasshouse?

Author Response

RESPONSES TO THE REFEREES´ COMMENTS

REVIEWER 1

Understanding interspecific reproductive barriers (IRBs) is a key concept in evolutionary biology and can also be informative for improving breeding. However, we currently know very little about which reproductive stage is driving reproductive isolation. The study does a great job at describing the pre- and post-zygotic IRBs present in Cucumis African wild species to give us an idea on the mechanisms that might be underlying speciation.

This is an exciting study, I have mostly minor suggestions for improvement of the manuscript. My main suggestion would be to integrate the significance of this research in the fields of evolutionary biology/plant breeding more throughout the manuscript. You touch on it in the final sentence of your abstract and the end of your discussion but I think your manuscript would improve from further development of the idea in your abstract, introduction and discussion.

Thank you very much for your kind comments. We have added a couple of statements in the introduction section to reinforce the significance of this research for plant breeding. This was indeed one of the final goals of this research line (lines 77-81 and 131-133).

Specific comments

Line 14 - Gap in knowledge needs to be made clear in the first couple of sentences of abstract.

Gap in knowledge has been addressed by adding a short sentence in lines 15-16 of the revised MS “However, nature of IRBs in Cucumis is largely unknown.”.

Line 89 – what is n referring to here?

 In this case ‘n’ means haploid chromosome number of the species. For the sake of clarity this has been changed in the revised MS by including the “chromosome number 2n” (lines 105-106).

Line 103 – which works don’t support Kirkbride? You initially said that paper on Line 94 supported Kirkbride.

We referred mainly to the last papers published on Cucumis phylogeny by Renner et al. 2007 [20], Ghebretinsae et al 2007 [21] and Sebastian et al. 2010 [2]. This has been corrected in the revised MS (line 122).

Line 133 – “(5-8)”.

 This has been corrected in the revised MS (line 160)

Line 134 – “(1-4)”.

 This has been corrected in the revised MS (line 161)

Line 134 – To improve clarity, add to Figure legend “Fertilized ovules were observed 24h after pollination” and change * to “Fertilized ovules were observed 48h after pollination”.

We did not observe fertilized ovules 48h post-pollination in the two crosses labelled with * in Figure 1(AxD and PxD) since, as stated in the Material and Methods section, “Pollinated pistils were collected 24h pp (just occasionally 48h pp) since we previously found pollen tubes reaching the ovules mostly before 24h in selfings and compatible crosses under our growing conditions.” (lines 586-588 original MS). However, following your suggestion to improve clarity we have added to the Figure´s 1 and 7 legend a new statement “Fertilized ovules were observed 24h after-pollination in all crosses with CI 1-4 (green color grade) except for those labelled with *” (lines 161-163 and 325-327 of the revised MS).

Line 144 – FIC is not significantly shorter based on the figure.

You are right, this has also been changed in the revised MS (line 173).

Line 146 – From the figure only MEL is significantly shorter. Please make it clear what is significantly different in this results section. Where are the ANOVA results for each comparison?

You are right, only MEL is significantly shorter and FIC name has been removed in the revised MS (line 173). In this context, ‘significantly different’ means that differences between means (ANOVA and Mann-Whitney-Wilcoxon tests) or mean ranks (Kruskal-Wallis) are significant according to the statistical analyses performed (see Materials and Methods section). These results are now included as Supplementary Table S2 in the ‘plants-2082980-supplementary’ file (lines 169-170 of the revised MS).

Line 149 – Why is it roughly?

“Roughly” has been replaced by “As main result we found that” in the revised MS (line 178).

Line 152 – These figures don’t look at the association between these variables so I wouldn’t refer to them here.

References to these figures have been removed the revised MS (line 181).

Line 170 – Fig. 2 – I found in Fig. 3 where you show where the sigma, style and ovary is very helpful to orientate myself, can you include this in Fig. 2 as well?

This modification has been performed in the Figure 2 of the revised MS.

Line 174 – Add what the yellow arrow heads are in the figure legend

This point has been corrected in the revised MS (lines 211-212).

Line 190 – Fig. 4 – Why have the names changed for the species? What are they now? It would be easier to follow the paper if the names were kept the same throughout.

You are right, Figure 4 was mistakenly labelled. This mistake has been corrected in the revised MS. Regarding species names, they have also been changed in this figure and, when it was necessary, throughout the revised MS. The following criteria were maintained: we use species abbreviations (first three letters) as indicated in the Supplementary Table S1 when referring to the species themselves or to their crosses. However, interspecific F1 hybrids are referred by the first letter of the parents separated by the symbol ‘×’ (also indicated in the Supplementary Table S1). For instance, P×D is the interspecific F1 hybrid derived from a cross between PUS and DIP and its backcross against the female parent is designed as (P×D)×PUS.

Line 192 – You need to refer to Fig. S2 here for fruit set %.

This has been corrected in the revised MS (lines 232-233).

Line 197 – Have you put the * in the wrong spot in Fig. 1? This line doesn’t match the figure currently.

We think the spots labelled with * in Figure 1 are correct since they correspond to the ANGxDIP and PUSxDIP crosses that were referred in lines 197-198 of the original MS “…in ANG×DIP and PUS×DIP interspecific crosses we did not observe pollen tubes reaching the ovules 24h post-pollination (pp)…”.

Line 198 – What do you mean by significant here?

You are right ‘significant’ was not the right term in this case. It was replaced by “reached 40% and 50%, respectively” in the revised MS (lines 236-237).

Line 205 – Fig. S2 - I think it would be clearer to have germination % in a different figure and to change species names in this figure to keep consistent with text (and most of paper).

This has been changed in the revised MS and now there is a different figure for germination % within Figure S2.

Line 213 – Explain how clustering was done here.

Clustering procedure is described in Material and Methods: “A hierarchical clustering heatmap representing post-zygotic IRBs in interspecific crosses (rows) was obtained on phenotypic data (columns: fruit set, fruit weight, seed number, seed weight and germination rate) by using the ClustVis web tool [61]. Heatmap rows and columns were clustered using correlation distance and average linkage. Unit variance scaling was applied to columns.” (lines 877-881 of the revised MS).

More in detail, it can be said that ClustVis web tool plots the heatmaps using pheatmap R package (version 0.7.7) and that package uses popular clustering distances and methods implemented in dist and hclust functions in R. In our case, we use correlation as the clustering distance (defined as Pearson´s correlation subtracted from1) and the average method as the linkage method.

Line 215 – on one side of what, explain to increase clarity.

This expression has been replaced by “as well as” in the revised MS (line 253).

Line 220 - Fig. 4 figure legend – explain parameters in order in which they appear in the figure. Also, explain what a value of 2 (red) and -1 (blue) mean and how the phylogeny was created.

Parameters are now explained in the order in which they appear in Figure 4 of the revised MS (lines 262-263).

Regarding the color key, heatmaps often show the Z-scores as the color key for each gene (in this case x variables) across samples (genotypes) where Z-score = x - mean (x)/standard deviation (x). The variable x measures the performance of a particular interspecific cross compared with the self-pollination of the seed parent. For instance, as for the fruit set in the cross DIPxZEY, x was calculated as follows: x (FS) = FSDXZ / FSDself so that, the closer is x to 0 the stronger are the IRBs for this cross.

We guess you meant clustering when you refer to phylogeny, if this is the case we tried to provide a more detailed explanation in the previous answer to Line 213 question.  

Line 234 – Fig. 5 figure legend. Add the fact that ‘parents’ and interspecific crosses are shown. Why is DIP parental stem length in FxD cross so much smaller than the other two?

Fig.5 legend of the revised MS now states that both interespecific hybrids and their parents are shown in the picture panels (lines 283-284).

The stem length (leaf pedicel) trait is also variable within the species (see Supplementary Table S2) and probably we did not select the most adequate DIP leaf for that picture. However, Supplementary Table S2 data show that DIP leaf pedicels are, on average, clearly longer than those of ZEY, PUS and FIC.

Line 252 - Fig 6. If you could order the bars by the order in which you have the hybrids it would be easier to find the two parents to compare the height of the bars.

This modification has been included in the Figure 6 of the revised MS.

Line 257 – Is PUS not one of the parents? Fix the writing here to increase clarity.

Yes, you are right, PUS is one of the parents. This mistake has been fixed in the revised MS (line 304).

Line 251 – where are these results?

These results have been incorporated in the revised MS as the Supplementary Table S4 Results show that direct and reciprocal crosses between the two used C. anguria accessions produce similar results than self-pollinations of the parents (in terms of fruit set, fruit weight, number of seeds per fruit and germination success). Furthermore, the obtained hybrids were male fertile and able to produce progeny after self-pollination.

Line 272 - Figure 7 – Keep CI index in same order as Figure 1 e.g. 1 to 8.

This has been modified in Figure 7 of the revised MS.

Line 313-314 – what do you mean by this sentence, clarify.

IRBs observed in Cucumis act at different moments in the life history (i.e. during pollen tube growth through the pistil, at the fertilization process or later on at the development of fruits and seeds) and these moments can be ordered chronologically and sequentially. Accordingly, the contribution of every reproduction stage to the reproductive isolation of the analyzed species may be quantified considering that every IRB can only avoid gene flow not removed by previous IRBs. As Ramsey et al (2003) state “Given a series of sequential stages of reproductive isolation, a reproductive barrier can only prevent gene flow that was not already eliminated by previous stages of isolation. Hence, components of reproductive isolation that act early in the life history contribute more to total isolation than barriers that function late”.

Line 326 - Table 1 – Add to table legend, how absolute and relative mean contribution to RI were calculated.

A footnote was added to Table 1 to clarify this point “Absolute and relative mean contribution to RI were calculated according to Ramsey et al. [23] (see Materials and Methods)” (lines 377-378).

Line 332 – Was ploidy in Table S1 based on flow cytometry? Add this to table legend.

This was corrected in the Supplementary Table S1 of the revised MS.

Line 337-341 – where is the results table for this?

These results were not included in the original MS but have been incorporated in the revised MS within the new Supplementary Table S5.

Line 354 – Briefly how is Nei’s genetic distance calculated?

Nei´s genetic distance [54] was calculated by means of “VCFR” [55] and “adegenet” [56] packages in R [57] using the set of 10967 GBS-derived SNPs filtered by TASSEL.

Line 383 - Figure 10 legend – you mean p>0.01? Usually done the other way round where significant values get *.

This point has been corrected in the Figure 10 of the revised MS and lines 435-436.

Line 410 - What do these results mean in a broader context?

According to Rieseberg and Blackman (2010), even if the cessation of gene flow between species is solely a consequence of geographical isolation it can be argued that speciation is not complete until genetically based barriers to gene flow evolve between geographically isolated populations and these barriers can arise at multiple prezygotic and postzygotic life-history stages. Therefore, our results suggest that IRBs in Cucumis did not arise in premating prezygotic stages but later on affecting the postmating pre- and postzygotic stages.

Line 488-489 – why oligogenic control?

Maybe ‘oligogenic control’ is not the most accurate expression. What we meant in this sentence is, in other words, that as the physiological requirements for pollen growth are expected to be similar between close species if two of these close species can be crossed in one direction but not in the reciprocal one (unilateral cross-incompatibility) not many genes are expected to be responsible for this difference. Text of these lines has been accordingly modified in the revised MS (lines 595-597).

Line 501 – what stage was more common for Nolana?

In Nolana postmating pre-zygotic isolation stages (i.e. pollen-pistil interactions) contribute only weakly to reproductive isolation (AC=0,07) while poszygotic stages (e.g. fruit set and others) are strong contributors (AC=0,76). This is in contrast with Cucumis, where contribution of both, pre- and postzygotic stages, is equivalent (AC=0,52 vs AC=0,47). To clarify this point we have replaced the corresponding statement by this one “However, in stark contrast with that observed in Nolana where pre-zygotic isolation was weak [41], pollen-pistil stage contribution to RI in Cucumis exceeded that of all post-zygotic stages.” (line 610 of the revised MS).

Line 563 – instead of glass greenhouse is it just a glasshouse?

This has been corrected in the revised MS (line 689).

Reviewer 2 Report

Investigating the mechanism of reproductive isolation is not only an essential topic of evolutionary biology but also has practical significance for crop breeding, as it helps to decide which wild species can be potentially used for the genetic improvement of crops. In this study, with the design of a full diallel cross, the authors tested the crossability among wild and cultivated Cucumis species during pre-mating, pre-zygotic and post-zygotic stages. By visualizing the development of pollen tubes, measuring fruit and seed sets, and examining F1 fertility, they found that both the pre- and post-zygotic barriers contribute to reproductive isolation among Cucumis species. However, the effect of the former one is slightly stronger. Their clustering analysis based on GBS-derived SNPs showed a positive correlation between the strength of isolation and genetic distance. More interesting is that unilateral cross-incongruity/incompatibility (UCI) was observed in some hybridization and backcrosses, and gene action analysis suggested that the pollen-pistil interaction and fruit set are mostly dominant.

Overall, this study was well designed and performed by making a mass of hybridizations and detailed morphological observations and combining multiple types of data. Their findings presented in the study have important implications for the breeding or even evolutionary studies of Cucumis species. I only have a few questions or suggestions for consideration.

-         What is the geographical range and habitat of these wild species? Have the ecogeographic isolation contributed to reproductive isolation in Cucumis? There is no need for adding analysis in this study, but including detailed information would be helpful to other studies.

-         Although the authors said there is a positive correlation between genetic distance and isolation, I wonder whether a trade-off between genetic incompatibility and heterosis existed in Cucumis, as suggested in Wei and Zhang’s paper on optimal mating distance (DOI: 10.1126/sciadv.aau55).

-         Due to the nature of phylogenetical sampling in this study, when the crossability or reproductive isolation was compared between individual crosses, they were not phylogenetically independent and violated the assumption of statistical independence. I suggest the authors make necessary corrections by, for example, phylogenetic mixed models (DOI: 10.1002/ece3.3093), which have been used to test hypotheses of reproductive isolation in the genus of Nolana and Silene.

Author Response

RESPONSES TO THE REFEREES´ COMMENTS

REVIEWER 2

Overall, this study was well designed and performed by making a mass of hybridizations and detailed morphological observations and combining multiple types of data. Their findings presented in the study have important implications for the breeding or even evolutionary studies of Cucumis species. I only have a few questions or suggestions for consideration.

Many thanks for your thoughtful and helpful comments.

What is the geographical range and habitat of these wild species? Have the ecogeographic isolation contributed to reproductive isolation in Cucumis? There is no need for adding analysis in this study, but including detailed information would be helpful to other studies.

Roughly, the Cucumis species studied in this work inhabit two distinct geographic areas. On the one hand, Cucumis anguria and C. zeyheri distribution areas overlap in Southeastern Africa where they coexist in the same ecoregions, though C. anguria can also be found in Southwestern Africa (e.g. Angola and Namibia). On the other hand, distribution areas of C. ficifolius, C. pustulatus, C. dipsaceus and C. melo overlap in Eastern central Africa, especially in the Somalia Massai ecoregion. These two groups seem to be separated from each other by the Afrotropical realm and the great Rift valley (Kirkbride, 1993; African Plant Database (version 4.0.0)) but, according to the available data, the existence of a contact zone between both groups in the Eastern South-Central Africa cannot be discarded. An overview of our data may suggest that crossability within these two groups is slightly better than between groups (e.g. in pollen-pistil and fruit set stages) therefore supporting a contribution of ecogeographic barriers to reproductive isolation. However, viable crossing is also possible between these two groups, so that even if geographic barriers have a role in reproductive isolation it may not be decisive. Unfortunately, our geographic data are not accurate enough to use them for further analysis. A summary of these comments has been included as part of the Discussion section in the revised MS (lines 449-455 of the revised MS).

Although the authors said there is a positive correlation between genetic distance and isolation, I wonder whether a trade-off between genetic incompatibility and heterosis existed in Cucumis, as suggested in Wei and Zhang’s paper on optimal mating distance (DOI: 10.1126/sciadv.aau55).

We have not extensive data on hybrid fitness-related traits. Anyway, if we focus on the leaf size, that can considered as a ‘component of fitness’, measurements seem to indicate that the three diploid interspecific hybrids obtained in this work (Zxa, PxD and DxZ) do not show heterosis (Supplementary Table S2 of the original MS). Nevertheless, they may show hybrid vigor for other fitness-related traits that we have not measured. Anyway, germination of hybrid seeds is compromised and all three hybrids are male-sterile so that genetic incompatibility effects could somehow offset the benefits of heterosis, if any.  

Due to the nature of phylogenetical sampling in this study, when the crossability or reproductive isolation was compared between individual crosses, they were not phylogenetically independent and violated the assumption of statistical independence. I suggest the authors make necessary corrections by, for example, phylogenetic mixed models (DOI: 10.1002/ece3.3093), which have been used to test hypotheses of reproductive isolation in the genus of Nolana and Silene.

Following your suggestion, and according to Castillo (2017), we have implemented a phylogenetic mixed model (PMM) by using the package MCMCglmm (Hadfield, 2010) to analyze our data. In particular, we use a similar approach to that described by Castillo (2017) for the Nolana dataset using a model that allows to incorporate multiple continuous variables with potential correlations including a pairwise genetic distance (GD) matrix. In this case, we used two potentially correlated variables (genetic distance and the pollen size/stigma+style length ratio) whose effects can be disentangled by using this model. Pollen size has been associated with the pollen provisioning capacity and therefore it may be related with its ability to grow through self- and non-self pistils. On the other way around, long stigma and/or styles may reduce the probabilities of non-self pollen tubes to reach the ovules. Thus, it could be speculated that the bigger is the pollen size/stigma+style length ratio for a given cross the lower is the reproductive isolation between two species.

However, according to the PMM model neither the GD nor the pollen size/stigma length ratio were significant predictors for RI at the pollen-pistil or the fruit set RI stages in Cucumis as can be seen in the Table below.

Coefficient

Biological meaning

Pollen – pistil interaction (prezygotic)

Fruit set (postzygotic)

µ (intercept)

Average RI

-0.3004

0.4663

-0.2710

0.6449

βgenetic distance

Slope relating genetic distance to RI

-0.2708

0.5371

-0.3625

0.5356

βmorphometry

Slope relating (pollen size / style + stigma length) to RI

-0.3774

0.4032

-0.3700

0.4465

This table shows a summary of coefficients estimated for the analysis of prezygotic (pollen -pistil interaction) and postzygotic (fruit set) reproductive isolation (RI) from the Cucumis data. The confidence intervals are for 95% highest posterior density (HPD) and are significant if they do not include zero.

These results are similar to that obtained by Castillo (2017) with the Nolana dataset where the PMM did not support GD and corolla diameter as significant predictors for RI. However, Jewell et al (2012) results show that correlation between GD and fruit set RI stage was significant according to the Mantel´s test.

In our case, no clear correlation was previously observed between floral attributes (pollen size and stigma+style length) and crossability. Therefore, it should be expected that PMM does not identify the pollen size/stigma+style length ratio as a significant predictor for RI. However, we did find a significant correlation between GD and fruit RI stage by the Mantel test (and with the pollen-pistil interaction and fruit set RI stages by Pearson´s test). Perhaps the strength of all these analyses might be limited by the lack of a wider representation of the Cucumis species. Nevertheless, according to the available data, we think that although the PMM does not support the tested variables as significant predictors for RI, the Mantel´s test results in Cucumis, as was the case with Nolana (Jewell et al. 2012), support the correlation between GD and RI. 

Round 2

Reviewer 2 Report

All my concerns have been addressed. 

Author Response

Many thanks again for your helpful comments and suggestions to improve the manuscript.